# StoRM : Stochastic Region Mixup

## Abstract

A number of data-augmentation strategies have been proposed to alleviate problems such as over-fitting, distribution shifts, and adversarial attacks in deep neural networks. A growing body of literature has investigated computationally expensive techniques like inclusion of saliency cues, diffusion processes or even fractal-like noise to improve upon robustness, clean accuracy. Although these methods may be intuitively compelling, there is limited theoretical justification for such techniques, especially given their computational inefficiencies and other issues. Thus, in this paper, we take a detour from them and propose Stochastic Region Mixup (StoRM). We simply focus on increasing the diversity of augmented samples. We show that this strategy can be extended to outperform saliency-based methods with lower computational overheads in several key metrics, and the key bottleneck in mixup based methods is the dimensionality of the vicinial risk space. StoRM—a stochastic extension of Region Mixup—stochastically combines multiple regions from a plurality of images leading to more diverse augmentations. We present empirical studies and theoretical analysis demonstrating that this richer augmentation space yields improved generalization and robustness while preserving label integrity through careful area-based mixing. Across benchmarks, StoRM consistently outperforms state-of-the-art mixup methods. The code will be released publicly upon acceptance.

## 1 Introduction

Data augmentation is a fundamental technique in deep learning that artificially expands datasets by introducing diverse transformations to existing samples. This approach enhances model generalization by exposing it to varied versions of the same data, fostering robustness with minimal computational overhead. However, conventional augmentation methods apply transformations independently to each image while preserving its label, which constrains their ability to extend beyond the natural data distribution. Consequently, they fall short in addressing challenges such as model overfitting and susceptibility to adversarial perturbations.

Image-mixing-based data augmentation methods offer a straightforward yet powerful approach to improving a model's ability to generalize to unseen data. These methods strategically blend randomly selected natural images and their corresponding labels from the training dataset using various mixing strategies, creating augmented images and labels. This process helps create more balanced class representations, reduces the likelihood of overconfident misclassifications, and refines decision boundaries, particularly in regions distant from the training data. Typically, such augmentation involves linear interpolation in either the input or feature space, to synthesize new training examples Their ability to improve deep learning model performance has been widely validated in various applications. Yet, a fundamental question remains open: *what exactly defines an effective interpolation between images?*

Several researchers (Kang & Kim, 2023; Uddin et al., 2021; Kim et al., 2020) have attempted to answer this question by proposing saliency-based mixup techniques, where the most informative regions of one image are superimposed onto the less critical areas (primarily the background) of another. Beyond saliency, other works have explored approaches such as diffusion-based processes (Islam et al., 2024) or the incorporation of fractal-like noise (Huang et al., 2023; Hendrycks et al., 2022).

However, these approaches come with significant computational overhead, making it challenging to achieve optimal performance efficiently. Beyond computational inefficiency, these methods are also constrained by the inherent limitations of saliency detection techniques, as well as by dependencies on intricate structure etc. The primary motivation behind this research direction is the argument that traditional mixup strategies produce overlayed images that appear unnatural (Yun et al., 2019) and may fail to preserve sufficient saliency information.

In this work, we take a detour and focus on increasing the diversity of augmented samples. Most mixup methods are limited to combining only two samples, with little analysis or design consideration for the impact of mixing more than two on model performance. Recent studies (Saha & Garain, 2024; Greenewald et al., 2023; Jeong et al., 2023) suggest that incorporating multiple samples can further enrich the diversity of augmented data. However, these approaches often achieve suboptimal performance due to restricted spatial configurations.

To address this issue, we introduce Stochastic Region Mixup (**StoRM**) to localize the blending operation in a stochastic manner while trying to retain saliency information. Rather than applying a single interpolation coefficient across the entire image, StoRM randomly partitions an image into tiles of varying sizes, and each tile is mixed with a different sample using its own Beta-distributed mixing coefficient. This localized blending not only preserves more spatial structure within each tile but also exposes the model to a much richer space of augmented samples. Crucially, StoRM weights label contributions by the tile's area, ensuring that mixed labels accurately reflect the proportion of each class in the composite image.

StoRM maintains the simplicity and computational efficiency of existing mixup-based approaches. Its implementation consists primarily of partition sampling and per-tile interpolation, which can be readily integrated into standard training pipelines without significantly increasing overhead. Through extensive experiments, StoRM demonstrates consistent improvements over strong baselines on a wide range of tasks, including general image classification, robustness, highlighting its broad applicability.

## 2 RELATED WORKS

Mixup (Zhang et al., 2018) creates augmented images by blending two randomly chosen images and their associated labels through linear interpolation. Manifold Mixup (Verma et al., 2019) promotes the learning of smooth interpolations between data points within the hidden layers of neural networks, resulting in enhanced accuracy compared to Mixup. CutMix (Yun et al., 2019) enhances performance by substituting a segment of an original image with a patch from a randomly chosen different image. Rather than selecting regions randomly, saliency can be used to identify objects from different images and combine them into a single image. SaliencyMix (Uddin et al., 2021) introduces a CutMix-inspired method, where a randomly sized patch is cut from the most salient region in an image. SnapMix (Huang et al., 2021) proposes an asymmetric replacement of a randomly sized patch in one image with a patch from another, guided by a class activation map (CAM) (Zhou et al., 2016) to align with semantic labels. PuzzleMix (Kim et al., 2020) transfers significant salient information to another image by solving an optimal transport problem. Co-Mixup (Kim et al., 2021) upgrades mixup by selecting diverse, suitable samples within a mini-batch, framing the mixing process as an optimization problem across multiple samples, and leveraging saliency maps from pre-trained models. GuidedMixup (Kang & Kim, 2023) introduces a greedy pairing algorithm to minimize the conflict of salient regions of paired images and achieve rich saliency in mixup images. However, saliency-based mixup methods incur significant computational costs to achieve optimal performance. AugMix (Hendrycks et al., 2020) applies a variety of transformations to generate highly diverse augmented images, strengthening corruption robustness and improving model calibration. To raise overall structural diversity, PixMix (Hendrycks et al., 2022) and IPMix (Huang et al., 2023) leverage synthetically generated images with intricate structures, like fractals. DiffuseMix (Islam et al., 2024) integrates a diffusion model with the mixup technique. AlignMixup (Venkataramanan et al., 2022) geometrically aligns two images in feature space, allowing interpolation between their features while keeping one image's spatial layout. This method retains the geometry or pose of one image and the texture of the other. Catch-up Mix (Kang et al., 2024) observed that CNNs favor powerful filters, so dropping slower-learning ones can hurt performance. It proposed a filtering module that merges features from weaker filters to yield mixed features with better

performance. $k$-Mixup (Greenewald et al., 2023) increased the diversity of augmented samples by interpolating data points based on the Wasserstein distance, introducing perturbations across batches of $k$ samples relative to additional $k$ reference points. DCutMix (Jeong et al., 2023) extended the conventional CutMix strategy by performing multiple cut-and-paste operations per sample. Empirical analyses demonstrated that such multi-sample augmentation techniques, particularly as explored in Landspace, guided neural networks toward broader (flatter) and deeper minima in the loss landscape. AutoMix (Liu et al., 2022) learns a patch-wise mixup policy through a lightweight "Mix Block" optimized jointly with the classifier — stabilized by a momentum pipeline — so it automatically generates label-consistent mixed images. Adversarial AutoMixup (Qin et al., 2024) alternately trains a mix-block to craft hard mixed images that maximize the classifier's loss and a classifier that learns to beat those adversarial mixes, yielding stronger regularization and robustness than standard AutoMix. The mixup literature is vast, so an exhaustive review is beyond our scope. Interested readers can find comprehensive overviews in the surveys by Cao et al. (2024); Jin et al. (2025).

## 3 STOCHASTIC REGION MIXUP

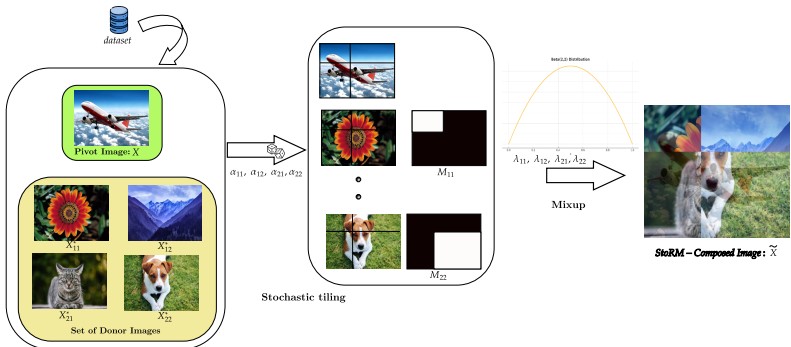

Figure 1: An overview of StoRM: Stochastic Region Mixup.

### 3.1 METHOD

Consider a training image $X \in \mathbb{R}^{W \times H \times C}$ with its one-hot encoded class label $y$. We aim to divide the image into $k \times k$ random-sized tiles by sampling $k-1$ partition points along the width and height axes. Let $P_w = \{p_0^w, p_1^w, \ldots, p_{k-1}^w, p_k^w\}$, where $p_0^w = 0$, $p_K^w = W$, and $0 < p_1^w < p_2^w < \cdots < p_{k-1}^w < W$, be the partition points along the width. Similarly, $P_h = \{p_0^h, p_1^h, \ldots, p_{k-1}^h, p_k^h\}$ with $p_0^h = 0$, $p_k^y = H$, and $0 < p_1^h < p_2^h < \cdots < p_{K-1}^h < H$ are the partition points along the height. The entire region is divided into $k \times k$ tiles $T_{ij}$, where $i, j \in [k]$. The set $\{1, \ldots, k\}$ is denoted as $[k]$. Each tile $T_{ij}$ is defined as

$$T_{ij} = \left\{ (w, h) \in [0, W) \times [0, H) \,\middle|\, \begin{array}{l} p_{i-1}^w \leq w < p_i^w, \\ p_{j-1}^h \leq h < p_j^h \end{array} \right\}. \tag{1}$$

This set represents all pixel coordinates $(w, h)$ within the boundaries defined by $p_{i-1}^w$ to $p_i^w$ and $p_{j-1}^h$ to $p_j^h$. For each tile $T_{ij}$, we define a binary mask $M_{ij} : [0, W) \times [0, H) \times [0, C) \to \{0, 1\}$ such that

$$M_{ij}(w, h, c) = \begin{cases} 1 & \text{if } (w, h) \in T_{ij}, \ c \in [0, C) \\ 0 & \text{otherwise} \end{cases}. \tag{2}$$

Thus, $M_{ij}$ equals one on all pixels (across all channels) belonging to tile $T_{ij}$. Let $X \in \mathbb{R}^{W \times H \times C}$ be an image that we want to distort and $\{X_1, X_2, \ldots, X_{k^2}\}$ be a set of $k^2$ images (each with labels $\{y_1, y_2, \ldots, y_{k^2}\}$) sampled from the distribution of images. We index them as

$$X_{ij}^* = X_{(i-1)k+j}, \quad y_{ij}^* = y_{(i-1)k+j}, \tag{3}$$

for $i, j \in [k]$. The mixed image $\tilde{X} \in \mathbb{R}^{W \times H \times C}$ is created by combining tiles from the original image $X$ with the corresponding tiling from the sampled images $X_{ij}^*$ using the mixing coefficient $\lambda_{ij}$:

$$\tilde{X} = \sum_{i=1}^{k} \sum_{j=1}^{k} \left[ M_{ij} \odot \left( \lambda_{ij} X + (1 - \lambda_{ij}) X_{ij}^* \right) \right]. \tag{4}$$

To ensure the labels reflect the same partition-based mixing, we define $\alpha_{ij}$ as the fraction of the total image area that tiles $T_{ij}$ occupies: $\alpha_{ij} = \dfrac{|T_{ij}|}{WH}$. Here, $|T_{ij}|$ denotes the (integer) number of pixels in tile $T_{ij}$. Then, the mixed label $\tilde{y}$ is given by

$$\tilde{y} = \sum_{i=1}^{k} \sum_{j=1}^{k} \left[ \alpha_{ij} \cdot \left( \lambda_{ij} y + (1 - \lambda_{ij}) y_{ij}^* \right) \right]. \tag{5}$$

Hence, each tile's contribution to the label is weighted by its spatial proportion $\alpha_{ij}$, and within that tile, we mix the labels of $X$ and $X_{ij}^*$ by the same $\lambda_{ij}$ used for the region mixing.

## 3.2 ALGORITHM

---

**Algorithm 1** $t$-th training iteration of StoRM

**Input:** Mini-batch $(\mathbf{X}, \mathbf{y})$ of size $N$, classifier $f$ with parameters $\theta_{t-1}$, model optimizer SGD
1: Sample $\lambda_{ij} \sim \text{Beta}(\beta, \beta)$ for $i, j \in [k]$
2: Sample $(k$-1) points independently from $\text{Uniform}(0, W)$ and sort them in ascending order:

$$0 < p_1^w < p_2^w < \cdots < p_{k-1}^w < W$$

3: Repeat the same for the height axis $[0, H)$ to obtain $0 < p_1^h < p_2^h < \cdots < p_{k-1}^h < H$
4: **for** $i = 1 \ldots k$ **do**
5:     **for** $j = 1 \ldots k$ **do**
6:         $(\mathbf{X}_{ij}^*, \mathbf{y}_{ij}^*) \leftarrow \textbf{RandomPermute}(\mathbf{X}, \mathbf{y})$ for
7:     **end for**
8: **end for**
9: Compute $(\tilde{\mathbf{X}}, \tilde{\mathbf{y}})$ using Equations (4) and (5)
10: $\mathcal{L} = \underbrace{\text{CE}\big(f(\mathbf{X}), \mathbf{y}\big)}_{\text{Standard Cross-entropy loss}} + \underbrace{\text{CE}\big(f(\tilde{\mathbf{X}}), \tilde{\mathbf{y}}\big)}_{\text{StoRM loss}}$
                                                                  $\triangleright$ *CE* is cross-entorpy loss.
11: $\theta_t \leftarrow \text{SGD}\left(\theta_{t-1}, \frac{\partial \mathcal{L}}{\partial \theta_{t-1}}\right)$
**Output:** Updated parameters $\theta_t$

---

Compared to standard mixup (Zhang et al., 2018), which typically merges two images using a single global mixing coefficient, *region mixup* (Saha & Garain, 2024) offers finer spatial control by segmenting the image into a fixed $k \times k$ grid and mixing each tile with a (potentially) different image (see Appendix A.1). This tile-wise approach mitigates the over-smoothing effect of a single $\lambda$ and produces more localized feature blends, thereby improving generalization beyond standard mixup. In addition, StoRM mixes up to $k^2$ images plus the original sample, assigning each tile an independent Beta-distributed coefficient $\lambda_{ij}$. This combination of stochastically placed tiles and per-tile mixing ratios generates a richer set of composite images, significantly enhancing the diversity of augmented samples.

However, region mixup still relies on a rigid grid partition that repeats identically across training samples, potentially failing to capture more varied spatial configurations. Stochastic Region Mixup addresses this limitation by randomly sampling partition boundaries for each image, thus mitigating the risk of repeatedly mixing the same contiguous blocks of pixels. Finally, by weighting label contributions proportionally to tile area, StoRM tries to maintain label integrity even when partitions vary in size, ensuring a more accurate alignment between image regions and their corresponding labels. Empirically, this fine-grained calibration (see Appendix A.2) of region-level mixing has

shown improvements in classification, robustness and other downstream tasks over existing mixup baselines in our experiments. We posit that these gains stem from StoRM exposing the network to a broader span of spatial configurations and class combinations. Figure 1 presents a schematic overview of the StoRM augmentation.

Random partitioning can juxtapose unrelated or contradictory semantic content, potentially confusing the model and hurting performance. Reducing semantic coherence may misguide models, particularly on datasets with highly structured spatial information. To mitigate this, following Saha & Garain (2024), we incorporate the standard cross-entropy loss as a regularization term in the StoRM loss, as described in Algorithm 1.

# 4 MECHANISTIC EXPLANATION: STORM AS A VICINAL KERNEL WITH ORTHOTOPIC GEOMETRY

Let $\mathcal{X} = \mathbb{R}^{W \times H \times C}$ and $\mathcal{Y} = \Delta^{m-1}$ be the input and label spaces. The dataset $D_n = \{(X_s, Y_s)\}_{s=1}^n$ is i.i.d. from an unknown distribution $P$ on $\mathcal{X} \times \mathcal{Y}$. For any measurable space $(\mathcal{Z}, \mathcal{A})$ and $z_0 \in \mathcal{Z}$, the *Dirac measure* $\delta_{z_0}$ is the probability measure $\delta_{z_0}(A) = \mathbf{1}\{z_0 \in A\}$ for $A \in \mathcal{A}$, equivalently $\int g \, d\delta_{z_0} = g(z_0)$ for bounded measurable $g$. The *empirical measure* is $\hat{P}_n = \frac{1}{n} \sum_{s=1}^n \delta_{(X_s, Y_s)}$. Given a predictor $f : \mathcal{X} \to \Delta^{m-1}$ and loss $\ell : \Delta^{m-1} \times \Delta^{m-1} \to \mathbb{R}_+$, ERM minimizes $\mathbb{E}_{(X,Y) \sim \hat{P}_n}[\ell(f(X), Y)] = \frac{1}{n} \sum_{s=1}^n \ell(f(X_s), Y_s)$. ERM trains on the empirical distribution $\hat{P}_n = \frac{1}{n} \sum_{i=1}^n \delta_{(X_i, Y_i)}$, which is a *pile of spikes* (point masses) at the training examples. Vicinal Risk Minimization (VRM) replaces each spike by a small *cloud* (a local neighborhood or "vicinity") around it via a vicinal kernel $K$, producing $v^{(n)} = \frac{1}{n} \sum_i K(\cdot \mid X_i, Y_i)$ (Chapelle et al., 2000).

**Definition 1** (Vicinal kernel and Vicinal Risk Minimization). *A vicinal kernel is a conditional probability kernel $K : (x, y) \mapsto K(\cdot \mid x, y) \in \mathcal{P}(\mathcal{X} \times \mathcal{Y})$ that prescribes a distribution of synthetic pairs $(\tilde{X}, \tilde{Y})$ in a neighborhood of $(x, y)$. The* vicinal distribution *associated with $D_n$ is $v^{(n)} = \frac{1}{n} \sum_{s=1}^n K(\cdot \mid X_s, Y_s)$, and the* vicinal risk *is $\mathcal{R}_v(f) = \mathbb{E}_{(\tilde{X}, \tilde{Y}) \sim v^{(n)}}[\ell(f(\tilde{X}), \tilde{Y})]$.*

Smoothing the empirical distribution into these local vicinities reduces effective complexity — intuitively, it removes the wiggle room a model has between isolated points. So, the hypothesis class *as seen through the data* is simpler (its Rademacher complexity under the vicinal distribution is smaller), which tightens generalization bounds and reduces overfitting. The price is a bit of bias — training on neighbors instead of the exact point — but when the kernel is local and respects the data geometry, that bias stays small while the variance reduction from averaging over vicinities is substantial.

**The StoRM vicinal kernel** Let $(X, Y)$ be an anchor sample. Let $\Pi$ denote the *random partition* of the image domain into $k \times k$ disjoint, axis-aligned tiles as specified in Section 3.1; a realization of $\Pi$ is the collection $\{T_{ij}\}_{i,j=1}^k$ with binary masks $M_{ij} \in \{0, 1\}^{W \times H \times C}$ (replicated across channels) and normalized areas $\alpha_{ij} := |T_{ij}|/(WH)$. Conditioned on $\Pi$, draw per-tile donors $\{(X_{ij}^*, Y_{ij}^*)\}$ from the training set and independent weights $\lambda_{ij} \sim \text{Beta}(\beta, \beta)$; write $t_{ij} := 1 - \lambda_{ij} \in [0, 1]$, $V_{ij} := M_{ij} \odot (X_{ij}^* - X)$, and $\Delta Y_{ij} := Y_{ij}^* - Y$. We algebraically rearrange the StoRM-generated vicinal pair

$$\tilde{X} = X + \sum_{i,j} t_{ij} V_{ij}, \qquad \tilde{Y} = Y + \sum_{i,j} \alpha_{ij} t_{ij} \Delta Y_{ij}. \qquad (6)$$

Randomizing $(\Pi, \{(X_{ij}^*, Y_{ij}^*)\}, \{\lambda_{ij}\})$ induces the conditional *vicinal kernel* $K_{\text{StoRM}}(A \mid X, Y) := \mathbb{P}\left((\tilde{X}, \tilde{Y}) \in A \mid X, Y\right)$, and the corresponding vicinal distribution over the dataset $v_{\text{StoRM}}^{(n)} = \frac{1}{n} \sum_{s=1}^n K_{\text{StoRM}}(\cdot \mid X_s, Y_s)$.

**Orthotopic geometry** We use the affine form in equation 6 (coefficients $\{t_{ij}\}$ and directions $\{V_{ij}\}$ determined by $\Pi$) to analyze the induced orthotopic geometry and smoothing.

**Lemma 1** (Disjoint masks $\Rightarrow$ orthogonality). *Let $D := WHC$ and equip $\mathbb{R}^D$ with the standard inner product $\langle A, B \rangle = \sum_{u=1}^D A_u B_u$. If $\text{supp}(M_{ij}) \cap \text{supp}(M_{i'j'}) = \varnothing$, then $\langle M_{ij} \odot A, \; M_{i'j'} \odot$*

$B \rangle = 0$, *for all* $A, B \in \mathbb{R}^D$. *In particular, for* $V_{ij} := M_{ij} \odot (X_{ij}^* - X)$ *as in equation 6, we have* $V_{ij} \perp V_{i'j'}$ *whenever* $(i, j) \neq (i', j')$; *hence any nonzero subfamily of* $\{V_{ij}\}$ *is linearly independent.*

> **Proposition 1** (StoRM vicinities are orthotopes). *Fix* $(X, \Pi, \{X_{ij}^*\})$ *and let* $t = (t_{ij}) \in [0,1]^{k^2}$. *The image set* $\mathcal{V}_{X,\Pi} := \left\{ X + \sum_{i,j} t_{ij} V_{ij} : t_{ij} \in [0,1] \right\}$ *is an orthotope* $\mathcal{O}(X; \{V_{ij}\}) \subset \mathbb{R}^D$ *with affine dimension* $\dim(\text{aff } \mathcal{V}_{X,\Pi}) = \text{rank}\{V_{ij}\} = \#\{(i,j) : V_{ij} \neq 0\} \leq k^2$, *and vertices* $\{ X + \sum_{i,j} \epsilon_{ij} V_{ij} : \epsilon_{ij} \in \{0,1\} \}$.

Proposition 1 identifies the StoRM vicinity (for fixed partition and donors) as a $d$-dimensional orthotope $\mathcal{V}_{X,\Pi}$ with $d = \#\{V_{ij} \neq 0\}$, which *strictly generalizes* global mixup: the classic 1D segment is recovered as a degenerate face when $k = 1$ or when all directions are collinear and the coefficients $t_{ij}$ are tied. Because tile masks are disjoint, these axes are orthogonal (Lemma 1), making the map $t \mapsto X + \sum t_{ij} V_{ij}$ block-separable; this structure removes first-order cross-terms and yields clean second-order analyses and interpretable regularization along semantically local directions. Across epochs, random partitions and donors generate a union of such orthotopes that broadens coverage without drifting far from the data manifold. StoRM preserves the spirit of mixup but lifts it from a single global direction to many orthogonal, part-level directions, delivering richer yet local vicinities and stronger, analyzable regularization that translate into improved robustness and generalization.

## 5 EXPERIMENTS AND RESULTS

In this section, we assess StoRM's performance and efficiency by contrasting it with baseline methods. We begin by examining its generalization capability on widely used classification benchmarks, including CIFAR-100 (Krizhevsky, 2009), CIFAR-10 (Krizhevsky, 2009), TinyImageNet (Le & Yang, 2015), and ImageNet (Deng et al., 2009). To further verify the breadth of its impact on generalization, we also evaluate StoRM on three Fine-Grained Vision Classification (FGVC) datasets—Caltech-UCSD Birds-200-2011 (CUB) (Wah et al., 2023), Stanford Cars (Cars) (Krause et al., 2013), and FGVC-Aircraft (Aircraft) (Maji et al., 2013). These datasets provide a highly diverse set of images, spanning flora, fauna, varied scenes, textures, transportation modes, human activities, satellite imagery, and general objects. Throughout this paper, we report experimental results assessing the effectiveness of StoRM across multiple architectures: ResNet (He et al., 2016a), PreActResNet (He et al., 2016b), DenseNet (Huang et al., 2017), and Wide-ResNet (Zagoruyko & Komodakis, 2016). We report performance metrics from previous research (Kim et al., 2020; Kang et al., 2024; Islam et al., 2024; Kang & Kim, 2023) for direct comparison.

### 5.1 GENERAL CLASSIFICATION

Table 1: Clean error rates for StoRM on CIFAR-10 ($\downarrow$). Each experiment reports the mean and standard deviation over three random seeds.

|  | Vanilla | MixUp | CutMix | AugMix | PixMix | IPMix | StoRM |
|---|---|---|---|---|---|---|---|
| WRN-28-10 | $3.8_{\pm 0.07}$ | $3.6_{\pm 0.08}$ | $3.4_{\pm 0.05}$ | $3.4_{\pm 0.07}$ | $3.8_{\pm 0.13}$ | $3.3_{\pm 0.08}$ | $\mathbf{3.02_{\pm 0.06}}$ |
| ResNet-18 | $4.4_{\pm 0.05}$ | $4.2_{\pm 0.04}$ | $4.0_{\pm 0.04}$ | $4.5_{\pm 0.03}$ | $4.4_{\pm 0.05}$ | $4.2_{\pm 0.07}$ | $\mathbf{3.86_{\pm 0.09}}$ |

**CIFAR-10** We conduct experiments on CIFAR-10 using two distinct backbone architectures: Wide ResNet-28-10 and ResNet-18. Our evaluation compares StoRM against IPMix and several other data augmentation techniques, including MixUp, CutMix, AugMix, and PixMix. Following the training protocol of IPMix (Huang et al., 2023), our results in Table 1 show that StoRM consistently achieves the highest accuracy across both architectures.

**CIFAR-100** Following different training protocols, we train two residual neural networks, WRN28-10 and PreActResNet18. For WRN28-10, we adhere to the IPMix training setup, while for PreActResNet18, we adopt the protocol from Verma et al. (2019) (Verma et al., 2019), training for 1200 epochs. Table 2 shows that StoRM achieves strong Top-1 and Top-5 accuracy across both architectures.

Table 2: Performance comparison of StoRM with state-of-the-art data augmentation methods on CIFAR-100.

| | PreActResNet-18 | | WRN-28-10 |
|---|---|---|---|
| | Top-1% | Top-5% | Top-1% |
| Vanilla | 76.33 | 91.02 | $81.0_{\pm 0.13}$ |
| MixUp | 76.84 | 92.42 | $81.6_{\pm 0.12}$ |
| Manifold | 79.02 | 93.37 | — |
| CutMix | 76.8 | 91.91 | $82.0_{\pm 0.11}$ |
| AugMix | 75.31 | 91.62 | $80.6_{\pm 0.11}$ |
| PixMix | 79.70 | — | $81.7_{\pm 0.13}$ |
| SaliencyMix | 79.75 | 94.71 | — |
| Guided-SR | 80.60 | 94.00 | — |
| Guided-AP | 81.40 | 94.88 | — |
| IPMix | — | — | $\mathbf{82.6_{\pm 0.25}}$ |
| PuzzleMix | 80.38 | 94.15 | — |
| Co-Mixup | 80.15 | — | — |
| **StoRM** | $\mathbf{82.06_{0.07}}$ | $\mathbf{95.55_{0.09}}$ | $\mathbf{82.6_{\pm 0.17}}$ |

**Tiny-ImageNet**  We train a PreActResNet18 network on the Tiny-ImageNet dataset, which consists of 200 classes, each containing 500 training images and 50 test images at a resolution of $64 \times 64$. Our training setup follows the PuzzleMix protocol. On Tiny-ImageNet, StoRM achieves significant improvements in classification performance. Compared to the Vanilla model, it yields notable Top-1 and Top-5 accuracy gains of $12.26\%$ and $13.58\%$, respectively. Furthermore, StoRM outperforms the second-best method, DiffuseMix, with additional gains of $3.72\%$ in Top-1 accuracy and $3.57\%$ in Top-5 accuracy. See Table 3 for a complete comparison.

Table 3: Top-1/Top-5 accuracy rates ($\%, \uparrow$) and FGSM error rate ($\%, \downarrow$) of mixup baselines trained on TinyImageNet datasets using PreActResNet-18. FGSM values are reported by averaging Kim et al. (2020); Kang et al. (2024).

| | Top-1%($\uparrow$) | Top-5%($\uparrow$) | FGSM(%,$\downarrow$) |
|---|---|---|---|
| Vanilla | 57.23 | 73.65 | 90.60 |
| MixUp | 56.99 | 73.02 | 89.85 |
| Manifold | 58.01 | 74.12 | 89.00 |
| CutMix | 56.67 | 73.59 | 87.10 |
| AugMix | 55.97 | 74.68 | 90.00 |
| SaliencyMix | 56.54 | 76.14 | 93.75 |
| Guided-SR | 59.44 | 75.54 | — |
| Co-Mixup | 64.15 | — | 91.11 |
| PuzzleMix | 63.48 | 75.52 | 89.87 |
| Guided-AP | 64.63 | 82.49 | — |
| DiffuseMix | 65.77 | 83.66 | — |
| **StoRM** | **69.49** | **87.23** | **82.84** |

**ImageNet**  Following the protocol outlined by Liu et al. (2022); Qin et al. (2024), we trained a ResNet-50 model on the ImageNet dataset for 100 epochs. As shown in Table 4, StoRM surpasses the next-best method, AdAutoMix (Qin et al., 2024), by $0.26\%$. It also improves on the vanilla implementation by $1.47\%$. Given this substantial difference, we believe StoRM provides a compelling trade-off, effectively balancing performance and efficiency—even on ImageNet. However, we do not compare with DiffuseMix (Islam et al., 2024), as it uses fractal blending and introduces significantly higher augmentation overhead.

Table 4: Top-1 and Top-5 accuracies comparison on ImageNet using ResNet-50. Compared numbers are taken from Islam et al. (2024); Liu et al. (2022); Qin et al. (2024).

| Method | Top-1 Acc. | Top-5 Acc. |
|---|---|---|
| Vanilla | 76.83 | 92.66 |
| AugMix | 76.75 | 93.30 |
| Manifold | 77.01 | 93.50 |
| Mixup | 77.12 | 93.52 |
| CutMix | 77.17 | 93.45 |
| SaliencyMix | 77.14 | — |
| Guided-SR | 77.20 | 93.66 |
| PixMix | 77.40 | — |
| PuzzleMix | 77.54 | 93.76 |
| GuidedMixup | 77.53 | 93.86 |
| Co-Mixup | 77.63 | 93.84 |
| AutoMix | 77.91 | — |
| AdAutoMix | 78.04 | — |
| **StoRM** | **78.30** | **94.16** |

## 5.2 ROBUSTNESS

**Robustness against Data Corruption**   we assess the model's resilience to data corruption using CIFAR-100-C, which introduces 19 different corruption types—such as snow, fog, blur, brightness changes, and noise—applied to the CIFAR-100 test set at five varying severity levels. To quantify robustness, we report the mean Corruption Error (mCE, %, ↓) of PreActResNet-18, averaged across all corruption types, as shown in Table 5. Our findings highlight that StoRM consistently outperforms other methods in handling corrupted data, achieving the lowest mCE. Notably, this improvement is achieved without any specialized training designed to counteract data corruption.

Table 5: Comparison of Mean Corruption Error (mCE, %, ↓) on CIFAR-100-C and FGSM error (%, ↓) rate on CIFAR-10 for PreActResNet-18.

| | FGSM Error(%, ↓) | mCE (%, ↓) |
|---|---|---|
| Vanilla | 84.93 | 51.43 |
| MixUp | **75.72** | 44.84 |
| Manifold | 81.96 | 44.32 |
| CutMix | 80.13 | 53.74 |
| SaliencyMix | 81.25 | 47.85 |
| Co-Mixup | 87.94 | 53.72 |
| PuzzleMix | 79.33 | 46.21 |
| **StoRM** | 76.04 | $\mathbf{43.11_{\pm 0.23}}$ |

**Robustness against Adversarial Attacks**   Here we examine the adversarial robustness of classifiers trained with different mixup-based augmentation strategies. We assess robustness by evaluating model performance on adversarial examples generated using the Fast Gradient Sign Method (FGSM) with an $l_\infty$ perturbation of $\frac{4}{255}$ applied to the test dataset. As shown in Table 2, StoRM significantly improves resistance to adversarial attacks on Tiny-ImageNet with PreActResNet-18, reducing the FGSM Top-1 error rate by $4.26\%$ compared to the strongest baseline. For CIFAR-100 (Table 5), StoRM maintains competitive robustness, achieving a Top-1 accuracy of $76.04\%$, closely aligning with MixUp's $75.72\%$.

## 5.3 AUGMENTATION OVERHEAD

Here, we assess both the computational efficiency and generalization performance of StoRM. To measure computational efficiency, we use the augmentation overhead, defined as the proportionate increase in training time due to the inclusion of augmentation. This metric is computed as follows

(Kang & Kim, 2023): *Augmentation Overhead* $= \dfrac{T_{aug} - T_{vanilla}}{T_{vanilla}} \times 100(\%)$, where $T_{aug}$ represents the total training time, while $T_{vanilla}$ refers to the training time without augmentation. To ensure a fair comparison, we measure the augmentation overhead without employing multi-processing during the augmentation process. Figure 2 demonstrates the effectiveness of our method, highlighting StoRM as a well-balanced solution that optimally trades off generalization performance and augmentation overhead.

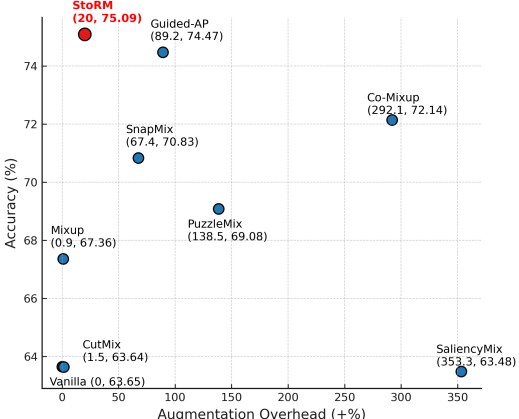

Figure 2: Augmentation overhead ($+\%$) vs. accuracy ($\%$) on the CUB dataset with a batch size of 16. The optimal augmentation strategies appear closer to the upper left corner, representing minimal overhead and higher accuracy.

Additional experiments on transfer learning, data scarcity, weakly-supervised object localization (WSOL), and ablation studies are provided in the Appendix C.

## 6 LIMITATIONS AND FUTURE WORKS

StoRM assumes that the objects within the images are distributed fairly evenly. If objects are highly localized (e.g., small objects in the center), mixing regions may result in excessive distortion and loss of meaningful features. Hyperparameters (number of partitions $k$, Beta-distribution parameter $\alpha$) could be crucial for adapting StoRM in datasets from different domains. Labels constructed using weighted averages of multiple samples might not adequately represent the semantic content of the mixed image. Furthermore, StoRM increases memory footprint as it requires loading multiple additional images (to execute the mixing process) per training instance. This can become a bottleneck for large-scale datasets, particularly when dealing with high-resolution images. While StoRM is primarily designed for images, its extension to other structured modalities, such as medical imaging, 3D point clouds, and video frames, holds significant potential. As future work, we will benchmark StoRM against Transformer-based architectures (e.g., ViT variants) (Chen et al., 2022; Li et al., 2022).

## 7 CONCLUSION

In this paper, we introduce Stochastic Region Mixup, a novel data augmentation technique that significantly increases both the diversity and effectiveness of mixed training samples for deep neural networks. Unlike conventional mixup methods, StoRM randomly divides images into variably sized tiles and applies region-specific mixing coefficients to each tile, balancing the need to preserve local saliency with the goal of maximizing diversity. By weighting label contributions in proportion to each tile's area, StoRM accurately represents the composite regions, leading to more effective training. Our findings demonstrate that key bottleneck in mixup based methods is the dimensionality of the vicinial risk space and enhancing the diversity of augmented data through stochastic and spatially nuanced mixing strategies can substantially improve model performance.

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
