## A APPENDIX

### A.1 PRELIMINARIES: REGION MIXUP

Region mixup (Saha & Garain, 2024) aims to generate a new training pair $(\tilde{X}, \tilde{y})$ by mixing regions from multiple training samples $(X_A, y_A), (X_{B_1}, y_{B_1}), (X_{B_2}, y_{B_2}), \ldots, (X_{B_{k^2}}, y_{B_{k^2}})$. The mixing process is described as follows:

$$\tilde{X} = \sum_{j=1}^{k^2} \lambda_j M_j \odot X_A + (1 - \lambda_j) M_j \odot X_{B_j}, \quad \text{and} \quad \tilde{y} = \frac{1}{k^2} \sum_{j=1}^{k^2} \lambda_j y_A + (1 - \lambda_j) y_{B_j}, \quad (7)$$

where $M_j \in \{0,1\}^{W \times H}$ is a binary mask indicating which region to mix from $X_A$ and $X_{B_j}$, ensuring that $\sum_{j=1}^{k^2} M_j = \mathbf{1}$. Here, $\odot$ represents element-wise multiplication. Setting $k = 1$ simplifies the approach to standard mixup regularization. As the region selection process, Region mixup divides an image evenly into $k \times k$ non-overlapping regions ($M_j$'s). This requires that the original image dimensions be divisible by $k$ to ensure all regions are of equal size. Mathematically,

$$M_j \in \left\{ \mathbb{1}_{B_{il}} \,\middle|\, i, l \in \{0, \ldots, k-1\} \right\}$$

where

$$B_{il} = \left[ \frac{iH}{k}, \frac{(i+1)H}{k} - 1 \right] \times \left[ \frac{lW}{k}, \frac{(l+1)W}{k} - 1 \right].$$

However, this fixed grid setup leads to suboptimal performance.

### A.2 LABEL CALIBRATION FOR CROSS-ENTROPY

When inputs are mixed, the targets must be mixed in the *same proportions*; with cross-entropy this is mathematically exact because the loss is *affine in the label*. Specifically, since $\ell_{CE}(p, y) = -y^\top \log p$ is linear in $y$, any convex combination of labels satisfies

$$\ell_{CE}\left(p, \sum_a w_a y^{(a)}\right) = \sum_a w_a \ell_{CE}(p, y^{(a)}).$$

In StoRM, the mixed label $\tilde{Y} = \sum_{i,j} \alpha_{ij}(\lambda_{ij} Y + (1 - \lambda_{ij}) Y_{ij}^*)$ is exactly such a convex combination with nonnegative coefficients $\{\alpha_{ij}\lambda_{ij}\} \cup \{\alpha_{ij}(1 - \lambda_{ij})\}$ that sum to 1, hence

$$\ell_{CE}(p, \tilde{Y}) = \sum_{i,j} \alpha_{ij}(\lambda_{ij} \ell_{CE}(p, Y) + (1 - \lambda_{ij}) \ell_{CE}(p, Y_{ij}^*)).$$

Thus the *same* tile weights used to mix the image also mix the label, guaranteeing that the loss on the mixed pair equals the corresponding mixture of losses on unmixed pairs (evaluated at the same prediction $p$). This label–input coherence prevents target mismatch, stabilizes gradients, and underpins the second-order analysis (first-order terms cancel in expectation); global mixup is the special case with a single weight $\lambda$ and no tiling .

## B THEORETICAL PROOFS

*Proof of the Lemma 1.* Since the supports are disjoint, $(M_{ij})_u (M_{i'j'})_u = 0$ for all coordinates $u$, whence $\langle M_{ij} \odot A, M_{i'j'} \odot B \rangle = \sum_{u=1}^D (M_{ij})_u A_u (M_{i'j'})_u B_u = \sum_{u=1}^D (M_{ij})_u (M_{i'j'})_u A_u B_u = 0$. If $\sum_{i,j} c_{ij} V_{ij} = 0$, taking the inner product with $V_{kl}$ yields $c_{kl} \|V_{kl}\|_2^2 = 0$, so $c_{kl} = 0$ for every $(k, l)$ with $V_{kl} \neq 0$. $\square$

*Proof of the Proposition 1.* Consider the affine map $\Phi : [0, 1]^{k^2} \to \mathbb{R}^D$, $\Phi(t) = X + \sum_{i,j} t_{ij} V_{ij}$. By Lemma 1, the supports of the $V_{ij}$'s are disjoint, so varying $t_{ij}$ changes only the coordinates inside tile $(i, j)$ and leaves all other coordinates fixed. Therefore, $\Phi$ decomposes as a *block-diagonal* map: the output coordinates split into $k^2$ disjoint blocks, the $(i, j)$-th block being the one-dimensional

segment $\{X_{\text{block}} + t_{ij}(V_{ij})_{\text{block}} : t_{ij} \in [0,1]\}$. The image of a Cartesian product of intervals under such a block-diagonal affine map is the Cartesian product of those intervals in disjoint coordinate blocks—an axis-aligned hyper-rectangle (orthotope). The vertex set of the orthotope in image space is exactly $\Phi(\{0,1\}^{k^2}) = \{X + \sum_{i,j} \epsilon_{ij} V_{ij} : \epsilon_{ij} \in \{0,1\}\}$. Finally, the affine dimension equals $\text{rank}\{V_{ij}\}$; by orthogonality, this rank is the number of nonzero $V_{ij}$'s. $\qquad\square$

## C ADDITIONAL EXPERIMENTS

### C.1 TRANSFER LEARNING

Transfer learning is a common strategy for adapting large architectures with limited resources and quick experimentation. While most image-mixing augmentation methods, except DiffuseMix Islam et al. (2024), lack evaluations in this setting, we evaluate StoRM in fine-tuning with Flower102, Aircraft, and Stanford Cars using an ImageNet-pretrained ResNet-50 from PyTorch, reporting results in Table 6.

Table 6: Top-1 (%) accuracy of **StoRM** on fine-tuning experiments with ResNet-50 pretrained on ImageNet.

| Method | Aircraft | Cars |
|---|---|---|
| Vanilla | 81.60 | 88.08 |
| DiffuseMix | 85.65 | 93.17 |
| **StoRM** | **89.58** | **93.48** |

StoRM achieved an accuracy of $89.58\%$ on the Aircraft dataset, outperforming DiffuseMix, which recorded $85.65\%$. A comparable pattern emerges in the Cars dataset, where StoRM attained $93.48\%$ accuracy. Given that fine-tuning is far more computationally efficient than training from scratch, these results underscore the practical benefits of StoRM.

### C.2 DATA SCARCITY

Deep neural networks are highly susceptible to overfitting when faced with data scarcity, limiting their ability to generalize effectively. With only a few training examples per class, these models often struggle to learn meaningful patterns. Since data augmentation is crucial in mitigating overfitting by expanding the effective training set, we investigate generalization performance under such constrained conditions. To explore this, we assess the performance of ResNet-18 on the Flower102 dataset Nilsback & Zisserman (2008) using just 10 images per class.

Table 7: Top-1 accuracy on the Flower102 dataset under data scarcity (10 images per class) using ResNet-18.

| Method | Validation Acc (%) | Test Acc (%) |
|---|---|---|
| Vanilla | 64.48 | 59.14 |
| Mixup | 70.55 | 66.81 |
| CutMix | 62.68 | 58.51 |
| SaliencyMix | 63.23 | 57.45 |
| SnapMix | 65.71 | 59.79 |
| PuzzleMix | 71.56 | 66.71 |
| Co-Mixup | 68.17 | 63.20 |
| Guided-SR | 72.84 | 69.31 |
| Guided-AP | 74.74 | 70.44 |
| DiffuseMix | 77.14 | 74.12 |
| **StoRM** | **79.51** | **76.80** |

As shown in Table 7, our proposed method, StoRM, outperforms other mixup-based augmentation techniques under this limited data condition, achieving a test accuracy of $76.80\%$ and a validation

accuracy of 79.51%. This demonstrates that StoRM enriches the training data diversity, strengthening the model's robustness in data-limited scenarios.

## C.3 WEAKLY-SUPERVISED OBJECT LOCALIZATION (WSOL)

Weakly Supervised Object Localization (WSOL) seeks to identify the location of an object in an image using only class labels, without relying on bounding boxes during training. It achieves this by capturing distinctive visual features that help the classifier concentrate on the most relevant regions of the image.

We train StoRM following the standard image classification procedure. During inference, as described in Venkataramanan et al. (2022), we generate a saliency map using Class Activation Mapping (CAM), apply a threshold of 0.15 to binarize it, and extract the bounding box from the resulting mask. Our experiments utilize a ResNet-50 model pretrained on ImageNet, which we fine-tune on the CUB200-2011 dataset. For evaluation, we adopt the methodology proposed in Choe et al. (2022), reporting localization performance via top-1 localization accuracy at an Intersection-over-Union (IoU) threshold of 0.5, as well as Maximal Box Accuracy (MaxBoxAcc-v2). We compare StoRM against the baseline CAM (without mixup), as well as Mixup, CutOut, CutMix. As shown in Table 8, StoRM surpasses Input Mixup, CutOut, and CutMix by 5.9%, 2.8%, and 0.4%, respectively.

Table 8: Weakly supervised object localization results on CUB200-2011. Top-1 Loc.: Top-1 localization accuracy (%,), MaxBoxAcc-v2: Maximum box accuracy. Higher values indicate better performance.

|  | Top-1 Loc. | MaxBoxAcc-v2 |
| --- | --- | --- |
| Baseline CAM | 49.4 | 59.7 |
| Input | 49.3 | 60.6 |
| CutMix | 54.8 | 64.8 |
| CutOut | 52.4 | — |
| StoRM | **55.2** | **65.2** |

## C.4 ABLATION STUDY

StoRM relies on two key hyperparameters—the tiling factor $k$ and the Beta-distribution concentration parameters $\beta$—which we hold constant across all experiments. To gauge sensitivity efficiently, we sweep each hyperparameter individually with a ResNet-18 classifier on CIFAR-10 and report the resulting Top-1 error rate. When varying $\beta \in \{0.1, 0.5, 1, 2, 5\}$, we fix $k = 2$; conversely, when varying $k \in \{1, 2, 3, 4\}$, we fix $\alpha = 2$.

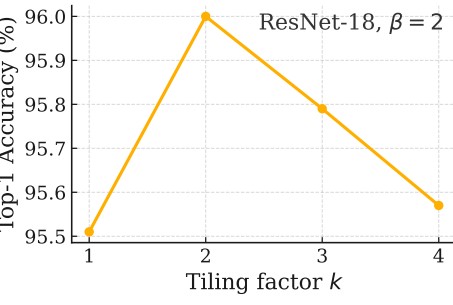 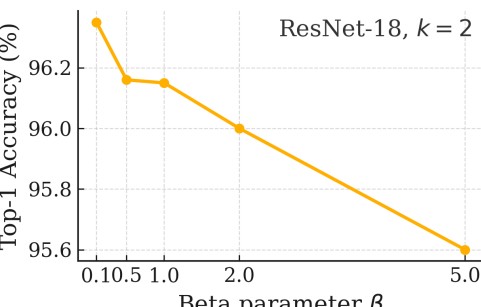

Figure 3: Ablation study on CIFAR-10 with ResNet-18. Left: accuracy as a function of the tiling factor $k$. Right: accuracy as a function of $\beta$ with $k = 2$.

**Tiling-factor sweep** ($\beta = 2$, **Fig. 3, left**)- Accuracy peaks at $k = 2$ (96.0%) and drops when the grid is either coarsened ($k = 1$) or made finer ($k = 3$–4). A $2 \times 2$ grid offers the best trade-off: each

tile is still large enough to preserve spatial context yet small enough to introduce beneficial diversity. Importantly, *all* multi-tile settings outperform the vanilla MixUp baseline ($k = 1$); even the finer grids with $k = 3$ and $k = 4$ reach $95.79\%$ and $95.57\%$, respectively, surpassing the single-tile accuracy of $95.51\%$.

$\beta$**-parameter sweep** ($k = 2$**, Fig. 3 Right**) – The model performs best at the smallest $\beta = 0.1$ (96.35 %), stays almost flat for $\beta \in \{0.5, 1, 2\}$, and drops noticeably at $\beta = 5$. A low $\beta$ skews the Beta distribution toward the end-points.