# OpenReview forum: "StoRM: Stochastic Region Mixup"
_ICLR.cc/2026/Conference — Submitted to ICLR 2026_

### Official Review · Reviewer_tTWV · 2025-10-14

**Soundness:** 2
**Presentation:** 4
**Contribution:** 1
**Rating:** 2
**Confidence:** 4

**Summary:**

This paper introduces StoRM, a stochastic extension of region-based MixUp. Empirically, StoRM achieves strong performance on CIFAR-10/100, Tiny-ImageNet, and ImageNet with little computational overhead.

**Strengths:**

Clear and well-written exposition. The algorithm is simple to understand and easy to reproduce.

Strong empirical results. StoRM performs competitively or better than many recent MixUp variants across multiple datasets and architectures.

Computational efficiency. Implementation requires only random partitioning and per-tile interpolation, making it lightweight compared to saliency-based or diffusion-based methods.

**Weaknesses:**

1. **Unclear theoretical contribution.**

   * The connection between the proposed *orthotopic vicinal kernel* and improved performance remains descriptive rather than explanatory.
   * The paper shows that StoRM defines a valid vicinal distribution, but **VRM theory alone does not predict why higher-dimensional orthotopes (larger k)** should yield better generalization. The analysis ends at geometric intuition without measurable quantities (e.g., Rademacher complexity, bias–variance trade-off, or margin bounds).

2. **Limited novelty relative to existing Mixup theory.**

   * Equation (6) reduces to standard Mixup when (k=1), and the same VRM interpretation already applies to MixUp, CutMix, or Manifold MixUp.
   * The claimed orthogonality of tile perturbations follows trivially from disjoint masks and does not clearly translate into a new theoretical insight.

3. **Empirical success lacks causal justification.**

   * While accuracy gains are consistent, it is unclear *why* stochastic partitioning helps. Could the same randomness be reproduced by simple multi-image MixUp or random cropping?
   * No ablation isolating the effect of stochastic vs. fixed partitioning, or the effect of k-value, is presented to support the theoretical claims.

**Minor concern**:  The method still increases memory footprint by requiring (k^2) donor images per sample.

**Questions:**

* Provide quantitative evidence linking the orthotopic dimension to effective generalization (e.g., plot accuracy vs. k or Rademacher complexity bounds).
* Compare against a baseline that mixes *multiple* images globally (k-MixUp) to isolate the benefit of local stochastic partitioning.
* Clarify whether the stochastic grid boundaries are resampled every iteration or fixed per epoch, since that could affect learning stability.
* Provide clear understanding why this can outperform "saliency" based method.

---

### Official Review · Reviewer_4rHL · 2025-10-16

**Soundness:** 3
**Presentation:** 3
**Contribution:** 2
**Rating:** 2
**Confidence:** 4

**Summary:**

This paper introduces a new mixup augmentation, **StoRM** (Stochastic Region Mixup). Unlike traditional Mixup or Region Mixup (e.g., CutMix, FMix), StoRM achieves finer-grained, spatially diverse mixing by **randomly partitioning image regions** and assigning each region an independent Beta-distributed mixing coefficient. Label contributions are weighted by region area to preserve semantic consistency. In the experimental section, StoRM undergoes systematic evaluation across multiple datasets (CIFAR-10/100, TinyImageNet, ImageNet, and FGVC). Results demonstrate superior performance over existing Mixup-based methods (including CutMix, PuzzleMix, Co-Mixup, IPMix, etc.) in classification accuracy, robustness (against noise and adversarial perturbations), and computational overhead.

**Strengths:**

* (S1) The paper proposes an augmentation framework combining “random region partitioning with independent mixing ratios,” significantly boosting data diversity without introducing additional model complexity. This represents a rational and concise extension of Region Mixup.

* (S2) Covering diverse tasks from small-scale (CIFAR) to large-scale (ImageNet) datasets, results demonstrate StoRM consistently outperforms comparable methods while showing clear advantages in robustness metrics (mCE, FGSM).

* (S3) The method relies solely on simple random partitioning and region-level interpolation operations, enabling direct integration into existing training pipelines with high practical applicability.

**Weaknesses:**

- **Lack of in-depth comparison and analysis of saliency-based methods**

   Have the authors researched saliency-region-based approaches? Why use random methods instead of guided ones?

- **Lack of performance demonstration for ViT-based models**

   The experiments lack generalization. The authors fail to demonstrate the performance of ViT-based methods (since they mention results from AdAutoMix and AutoMix, they should note that both conducted experiments on ViT methods, e.g., DeiT-Small, Swin Transformer-Tiny, and ConvNeXt-Tiny).

- **Lack of large-scale experiments**

   The authors appear to have selected only models with small parameters. Are there results for ResNeXt50 or other model with lager parameters? Authors can refer to the experimental results in the Openmixup [1] repository.

- **Typo**

   Where are the DenseNet results? I couldn't find them in this **main paper** or the **supplementary materials**.

[1] Openmixup: Open mixup toolbox and benchmark for visual representation learning[J]. arXiv preprint

**Questions:**

Seeing weaknesses. If the author can solve this concern of mine, I will consider raising my score.

---

### Official Review · Reviewer_PD6i · 2025-10-16

**Soundness:** 3
**Presentation:** 2
**Contribution:** 2
**Rating:** 2
**Confidence:** 5

**Summary:**

The author proposed a local, multi-image mixup by randomly partitioning an image into k×k tiles and blending each tile with a different sample via independent Beta coefficients, weighting labels by tile area.

**Strengths:**

1) Empirically, StoRM achieves top performance on benchmarks: e.g. on TinyImageNet it attains 69.49% Top-1 (vs. 65.77% for previous best DiffuseMix) and 82.84% FGSM accuracy; on ImageNet it reaches 78.30% (vs. 78.04% AdAutoMix).
2) StoRM also yields strong robustness: it has the lowest CIFAR-100-C mean corruption error (43.11% vs. MixUp’s 44.84%), and cuts TinyImageNet FGSM error by ≈4.3% relative to the strongest baseline.

**Weaknesses:**

1) The references are missing in Introduction section, I will request author to read previous SOTA paper for better reference understanding.
2) SnapMix and AlignMixup are cited but not evaluated; the CIFAR-10 FGSM table shows StoRM’s error (76.04%) is only marginally higher than MixUp’s (75.72%).
3) StoRM fixes k=2 and β=2 for all experiments, with a brief sweep on CIFAR-10 indicating k=2 is optimal; wider hyperparameter studies (different k, β, or number of samples) are not shown.
4)  StoRM requires sampling k^2 images per example, increasing memory; the authors claim low overhead (Fig.2) but provide no quantitative comparison.


Missing References

SUMix: Mixup with Semantic and Uncertain Information

Context-guided Responsible Data Augmentation with Diffusion Models

GenMix: Effective Data Augmentation with Generative Diffusion Model Image Editing

Effective Data Augmentation With Diffusion Models

**Questions:**

1) Author should focus on introduction section and consistent comparison for instance divide your table into two sections simple mixup methods and generative mixup methods
2) I would prefer the author should upload the paper on arxiv in this the idea remain valid.

---

### Official Review · Reviewer_WTXg · 2025-10-17

**Soundness:** 2
**Presentation:** 2
**Contribution:** 2
**Rating:** 4
**Confidence:** 4

**Summary:**

This paper proposes StoRM, a stochastic region-based data augmentation method designed to improve model accuracy by increasing the diversity of augmented samples. Experiments on CIFAR-10, CIFAR-100, and ImageNet show that the proposed method achieves strong performance and enhances generalization compared to existing Mixup-based approaches.

**Strengths:**

1. Practical improvement: StoRM demonstrates consistent empirical gains across multiple benchmarks (CIFAR-10/100, TinyImageNet, ImageNet) and various architectures.

2. Comprehensive evaluation: The paper includes robustness tests (adversarial attacks, corruption), fine-grained classification, and computational overhead analysis.

**Weaknesses:**

1. Limited novelty: The core idea is essentially a stochastic extension of Region Mixup. The main differences are (i) random partition boundaries instead of a fixed grid, and (ii) independent λ values per tile rather than a shared coefficient. These represent incremental modifications rather than a fundamental methodological innovation.

2. Unclear contribution decomposition: The paper lacks ablation studies that isolate the effects of each component (e.g., random partitioning, per-tile λ sampling, stochastic ratio). This makes it difficult to assess which design choices are primarily responsible for the observed improvements.

3. Marginal gains on some datasets: On CIFAR-10/100 with WRN-28-10, the improvements are within noise margins (≈0.3% on CIFAR-10 and on par with IPMix on CIFAR-100), raising questions about the practical significance of the method’s benefits.

**Questions:**

1. Can you provide a proper ablation study decomposing your contributions?

2. What is the computational overhead of StoRM in terms of FLOPs and memory usage compared to standard Mixup or CutMix?

3. Can you show qualitative examples of generated augmentations?

4. Have you tried this on Vision Transformers as claimed in limitations?

---

### Meta-Review · Area_Chair_Q3KB · 2026-01-06

**Summary:**

This paper proposes an image augmentation strategy that increases the diversity of augmented samples. The summarized reviewers' concerns are:

1. Lack of novelty of the proposed strategy compared to the previously proposed image-based data augmentation.
2. Weak experiments with limited comparison methods, ablation study, and architectural diversity.
3. Marginal performance gain.

The authors have not submitted their rebuttal, so the above concerns have not addressed at all.
Therefore, the AC's decision for this paper is rejection.

**Reviewer Concerns:**

There was no rebuttal, so all the concerns remain.

**Reviewer Scores:**

There was no rebuttal, so I think the reviewers' scores remain.

---

### Decision · Program_Chairs · 2026-01-26

Reject